# Chiroptical spectroscopy of a freely diffusing single nanoparticle

Johannes Sachs[1,2,4], Jan-Philipp Günther [1,2,4], Andrew G. Mark[1,3] & Peer Fischer [1,2✉]

Chiral plasmonic nanoparticles can exhibit strong chiroptical signals compared to the corresponding molecular response. Observations are, however, generally restricted to measurements on stationary single particles with a fixed orientation, which complicates the spectral analysis. Here, we report the spectroscopic observation of a freely diffusing single chiral nanoparticle in solution. By acquiring time-resolved circular differential scattering signals we show that the spectral interpretation is significantly simplified. We experimentally demonstrate the equivalence between time-averaged chiral spectra observed for an individual nanostructure and the corresponding ensemble spectra, and thereby demonstrate the ergodic principle for chiroptical spectroscopy. We also show how it is possible for an achiral particle to yield an instantaneous chiroptical response, whereas the time-averaged signals are an unequivocal measure of chirality. Time-resolved chiroptical spectroscopy on a freely moving chiral nanoparticle advances the field of single-particle spectroscopy, and is a means to obtain the true signature of the nanoparticle's chirality.

---

[1] Max Planck Institute for Intelligent Systems, Heisenbergstr. 3, Stuttgart 70569, Germany. [2] Institute of Physical Chemistry, University of Stuttgart, Pfaffenwaldring 55, Stuttgart 70569, Germany. [3]Present address: Metamaterial Inc., Dartmouth, NS B2Y 4M9, Canada. [4]These authors contributed equally: Johannes Sachs, Jan-Philipp Günther. ✉email: fischer@is.mpg.de

Chirality, also known as handedness, is the property that describes the geometrical difference between an object and its mirror image. It is an important symmetry property exhibited by almost all biomolecules and is seen as a signature of life on Earth[1]. The presence of chirality in nanostructures is associated with unique optical polarization properties. Optically resonant chiral nanostructures may offer a means to enhance weak optical signals associated with molecular chirality[2–4]. The power of all spectroscopic probes of chirality lies in their ability to reveal subtle geometric details[5]. Chiroptical spectroscopy is based on the differential interaction of left- and right-circularly polarized light with the sample. A change in the phase of circularly polarized light gives rise to optical rotation due to circular birefringence (CB)[6]. A difference in absorption (strictly, extinction) is known as circular dichroism (CD). Both are unequivocal signatures of the presence of chirality when observed from an isotropic solution or suspension.

Chiral spectroscopy clasically uses solutions containing a large number of randomly (isotropically) oriented chiral molecules or particles[7]. However, ensemble measurements not only require large sample quantities but also cause the loss of important spectral information, which has been well-recognized in the field of single-molecule spectroscopy[8]. Especially information on dynamic processes and variations within the analyte distribution are invariably lost by ensemble averaging. In contrast, single-particle measurements only need one particle to provide rich information as has been demonstrated, for instance, by orientation-sensing of single Au nanorods with linearly polarized light[9,10]. In recent years, single nanoparticles and their differential interaction with circularly polarized light has been studied with darkfield microscopy[11–14] and luminescence[15]. In these measurements the signal for left- and right-circularly polarized light is collected in two separate experiments. In addition, the examined particle remains stationary (immobilized to a surface) and therefore oriented with respect to the input light. In this case contributions from linear dichroism (LD) and linear birefringence (LB) can cause nonzero circular differential intensities[16], which are exacerbated by coupling effects with the substrate[17] or the optics[12,13]. Consequently, even achiral particles exhibit chiroptical responses, which do not represent the intrinsic chirality of the single nanoparticle, i.e., they are by no means equivalent to classical ensemble CD responses. A fact that has long since been recognized for molecular CD spectroscopy[18,19] and that has thus far hampered the observation of a chiral spectrum from a single molecule[20,21].

Here, we observe the circular differential scattering intensity (CDSI) spectrum of a single freely diffusing Brownian nanoparticle. We demonstrate an observable that, to the best of our knowledge, has not been reported previously—the 'true' chiroptical spectrum of a single nanoparticle, which unequivocally determines its intrinsic chirality. The difficulties arising from oriented nanostructures are avoided by recording both the left- and right-circularly polarized scattered light simultaneously and continuously in a single experiment while the particle's Brownian motion causes its random reorientation. We demonstrate that the time-series of CDSI spectra (see Fig. 1a) can be used to unequivocally determine the chirality of a single nanoparticle in solution. In accord with the ergodic principle, linear optical anisotropies vanish for achiral structures. Ergodicity and Brownian motion therefore permit us to record a chiral spectrum from a single nanoparticle, which is equivalent to classical ensemble spectroscopy. Deducing the same information from a single plasmonic nanoparticle as from traditional CD spectroscopy enables novel sensing techniques with strongly enhanced sensitivities.

## Results

**Chiral spectroscopy and the ergodic hypothesis.** The employed single-particle spectrometer is based on a balanced detection scheme[22], which we have extended to record full spectra. This permits us to simultaneously capture left- and right-circularly polarized scattered light, $I_L$ and $I_R$, on a single detector (see Fig. 1c). Briefly, a dark-field condenser is used to illuminate a single nanoparticle with a hollow cone of unpolarized light. The light scattered off the particle is collected and redirected to polarization optics that comprises two quarter-waveplates (QWP) and a Wollaston prism (WP), which spatially separate the intensities $I_L$ and $I_R$. A spectrograph then disperses both beams before they are collected on the same detector. By subtracting and

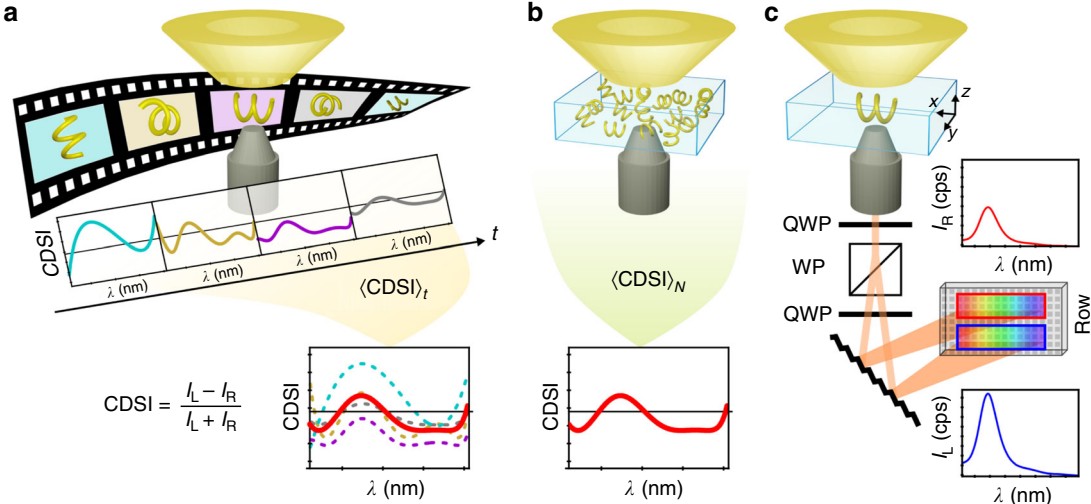

$$CDSI = \frac{I_L - I_R}{I_L + I_R}$$

**Fig. 1 Single particle chiroptical spectroscopy versus ensemble spectroscopy. a** A time-series experiment acquires snapshots of the circular differential scattering intensity (CDSI) at arbitrary instants of time at which the particle has a fixed orientation in space and which yields the time-averaged $\langle CDSI \rangle_t$. **b** An ensemble-average $\langle CDSI \rangle_N$ originates from $N$ individual particles, which are isotropically oriented in solution. **c** Schematic of the setup, which utilizes a dark-field condenser for directing the incident light onto the sample and a microscope objective to collect the light scattered off the particle. The objective is followed by polarization optics, consisting of two quarter-waveplates (QWP) and a Wollaston prism (WP) to simultaneously acquire spectra of scattered left- and right-circularly polarized light (LCP and RCP) on a single detector array.

normalizing the two spectra we obtain the CDSI defined as[23]:

$$\mathrm{CDSI} = \frac{I_\mathrm{L} - I_\mathrm{R}}{I_\mathrm{L} + I_\mathrm{R}}, \qquad (1)$$

which is in this configuration equivalent to the degree of circular polarization of the scattered light as is confirmed by the Mueller-Stokes formalism (see Supplementary Note 5).

In our setup, a spectrum of a single particle was acquired at one specific instant in time, i.e., with an exposure time short enough to assume a quasi-stationary orientation in 3D space during the acquisition of one spectrum (see "Methods" and Supplementary Note 1). If the particle moves and if the observation is repeated with the same particle at several instants of time, the measurement will yield a time-series of spectra originating from distinct orientations (Fig. 1a). According to the ergodic hypothesis the time average of any observable, here CDSI, over a sufficiently long period ($\langle\mathrm{CDSI}\rangle_t$) will yield the same result as an ensemble measurement of a sufficiently large number $N$ of identical systems (here identical copies of the same particle), such that $\langle\mathrm{CDSI}\rangle_t = \langle\mathrm{CDSI}\rangle_N$. The probability distribution describing the particle's orientation must be the same in both cases[24], which holds for our experiments, since the particles undergo the same statistical reorientation due to Brownian motion.

**Spectroscopy of a single freely rotating chiral nanohelix.** To demonstrate that the time-series of CDSI spectra measures chirality intrinsic to a single particle, we prepared Au nanohelices

with a recently extended physical vapour deposition technique[25] based on glancing angle deposition[26]. Afterwards the helices (scanning electron micrographs (SEM) in Fig. 2a) are transferred into solution. In order to study the nanoparticles in liquid we immersed them in a viscous water–glycerol mixture. This reduced their translational Brownian motion significantly, while ensuring that the nanoparticles' rotational motion remained high enough to sample isotropic orientations over the course of the time-series experiment. We recorded a full spectrum corresponding to one snapshot/frame depicted in Fig. 1a in 1 s. The particles' rotational motion is slow enough that in one frame not all orientations are isotropically sampled (see "Methods" and Supplementary Note 1). In Fig. 2b the CDSI of a freely diffusing left-handed helix is shown at 750 nm, averaged over a spectral width of 10 nm ($\lambda = 750 \pm 5$ nm), for the first 60 frames of a time-series. It is readily seen that the measured CDSI changes from frame to frame. Even changes in sign are observed. The same is true not only for a single wavelength but also for the full spectrum as can be seen in Fig. 2c, where the first 30 frames of the same measurement as in Fig. 2b are plotted. For certain times (orientations) the CDSI spectrum appears positive overall while on average it is negative. This observation shows that snapshots of a nanoparticle with a fixed orientation do not provide a measure of its true chirality. Deducing the presence or absence of chirality from a single spectrum at one orientation is thus in general not possible. However, the time-averaged signal $\langle\mathrm{CDSI}\rangle_t$, which probes all orientations, yields a true signature of a single particle's chirality.

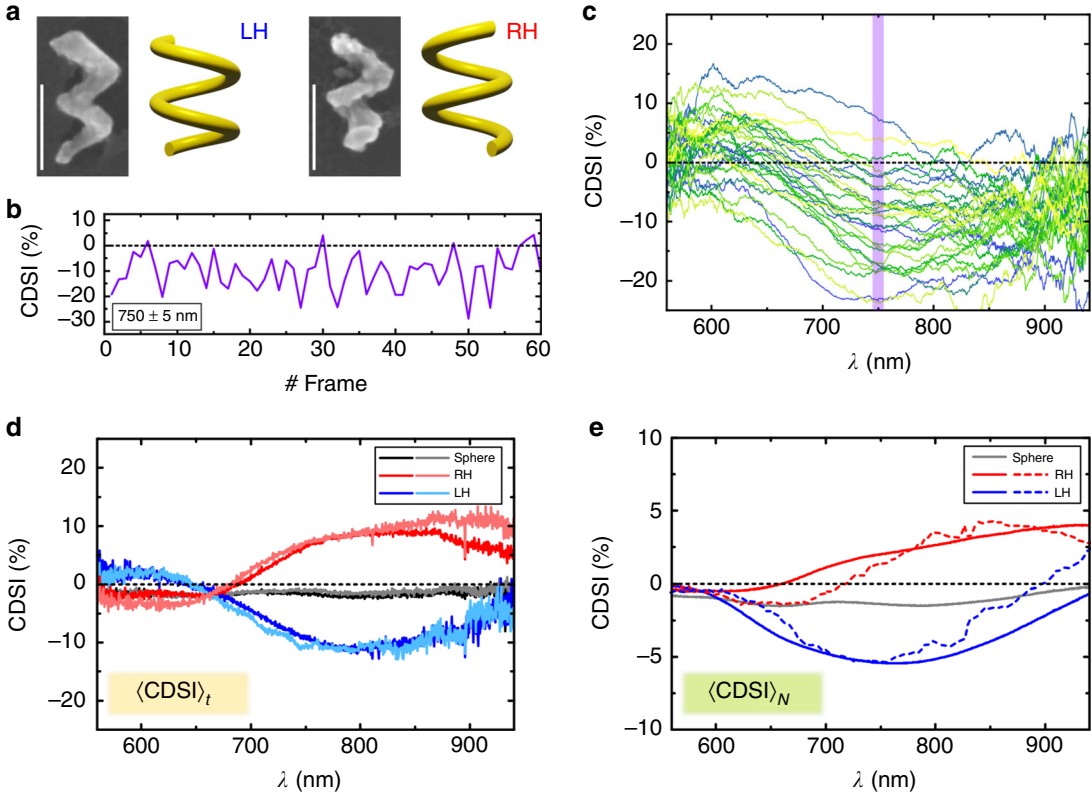

**Fig. 2 Time-resolved chiroptical spectroscopy of freely diffusing single nanohelices. a** SEM and schematics of Au nanohelices (scale bar 100 nm). **b** Frame-wise CDSI at $\lambda = 750 \pm 5$ nm for a single left-handed nanohelix that is undergoing Brownian motion. **c** Full CDSI spectra for momentarily aligned orientations in 3D space corresponding to the first 30 frames shown in (**b**). The purple vertical line indicates the spectral region shown as a time-trace in (**b**). **d** $\langle CDSI\rangle_t$ ($t = 400$ frames) for freely diffusing single Au nanohelices and nanospheres. For each shape two measurements from different particles are shown in different shades of grey, red and blue. **e** $\langle CDSI\rangle_N$ of the same Au helices and spheres presented in **d** measured in the setup according to Fig. 1b (solid lines) and by utilizing a classical chiroptical measurement with a cuvette (dashed lines, data scaled). The latter detects under a different scattering angle (see Supplementary Fig. 3 and Note 2).

$\langle CDSI \rangle_t$ is shown in Fig. 2d for left- (LH) and right-handed (RH) Au nanohelices ($t = 400$ frames), as well as Au spheres ($t = 60$ frames). As expected, the time-averaged CDSI of a sphere vanishes, whereas the two helices with opposite handedness exhibit signals of opposite sign. Notice that lower viscosities would speed up rotational diffusion and consequently also the measurement time to obtain an isotropic orientation sampling (see "Methods").

In contrast to molecules, it is not feasible to observe a pure ensemble of identical nanoparticles due to unavoidable variations and impurities that arise during their fabrication. It follows that the time-averaged CDSI of a single nanoparticle, to our knowledge, provides an otherwise not yet accessible observable—the unique chiroptical spectrum of a single particle without additional contributions due to linear polarizations. However, for comparison we also measured an ensemble of particles ($\langle CDSI \rangle_N$) in the same setup. For this, the scattered light was collected over a large field of view and for a dense colloidal solution as schematically depicted in Fig. 1b. This ensured that tens of particles contribute to a reasonable ensemble average (see "Methods" and Supplementary Fig. 4 and Note 3). Further, the ensembles were also measured in a commercial CD spectrometer utilizing a cuvette, which detects under a different scattering angle compared to the dark-field spectrometer. Consequently, measuring the CDSI of an ensemble with our setup is closely related but not equal to an ensemble measurement in a CD spectrometer, because the detected intensities in general depend on the scattering angle[27] (see Supplementary Fig. 3 and Supplementary Note 2). Both kinds of ensemble-averaged spectra $\langle CDSI \rangle_N$ are shown in Fig. 2e. Spheres show no CDSI, as expected for achiral particles, whereas the helices exhibit spectra of opposite polarity. The peak maxima are shifted because the ensemble is now immersed in a medium (water) with different refractive index.

The data are in excellent agreement with the time-averaged spectra $\langle CDSI \rangle_t$ in Fig. 2d, indicating the ergodicity of the system despite the shape variations within the ensemble.

**Spectroscopy of single freely rotating achiral nanostructures.** For molecules it is well-known from the theory of optical activity[18] and the measurement of CD[21] that an oriented sample, including molecules immobilized at an interface, displays spectra that significantly differ from rotationally averaged (ensemble) CD spectra. In the latter case, contributions from electric-quadrupole transitions average to zero, whereas those transitions have to be considered for a stationary aligned structure. Similarly, contributions due to any LB or LD in an (oriented) sample can alter the polarization state and contribute to the circular intensity difference signal[19], which does not reflect the chirality intrinsic to the molecule. For plasmonic samples orientation-dependent effects have been recognized[11,28], although it is possible to partially mitigate them with a carefully designed sample[14,29,30], which limits such studies to highly symmetrical structures.

Anisotropic single plasmonic particles, like nanorods, can exhibit strong LD[9,10] and hence, because of the Kramers–Kronig relation, also LB. An achiral nanorod can thus display polarization and intensity fluctuations as its orientation changes, which will in turn be detected as a nonzero CDSI, even though the structure is intrinsically achiral. Small imperfections of the dark-field condenser will exacerbate such influences as has been shown in previous work[12,13]. To demonstrate that single achiral nanoparticles can yield misleading chiroptical spectra due to linear polarizations effects, we observe spectra of nanorods and -spheres that are randomly moved by thermal noise.

First, we discuss the time-series experiment on Au nanorods prepared by PVD (SEM in Fig. 3a). The resonance of a single

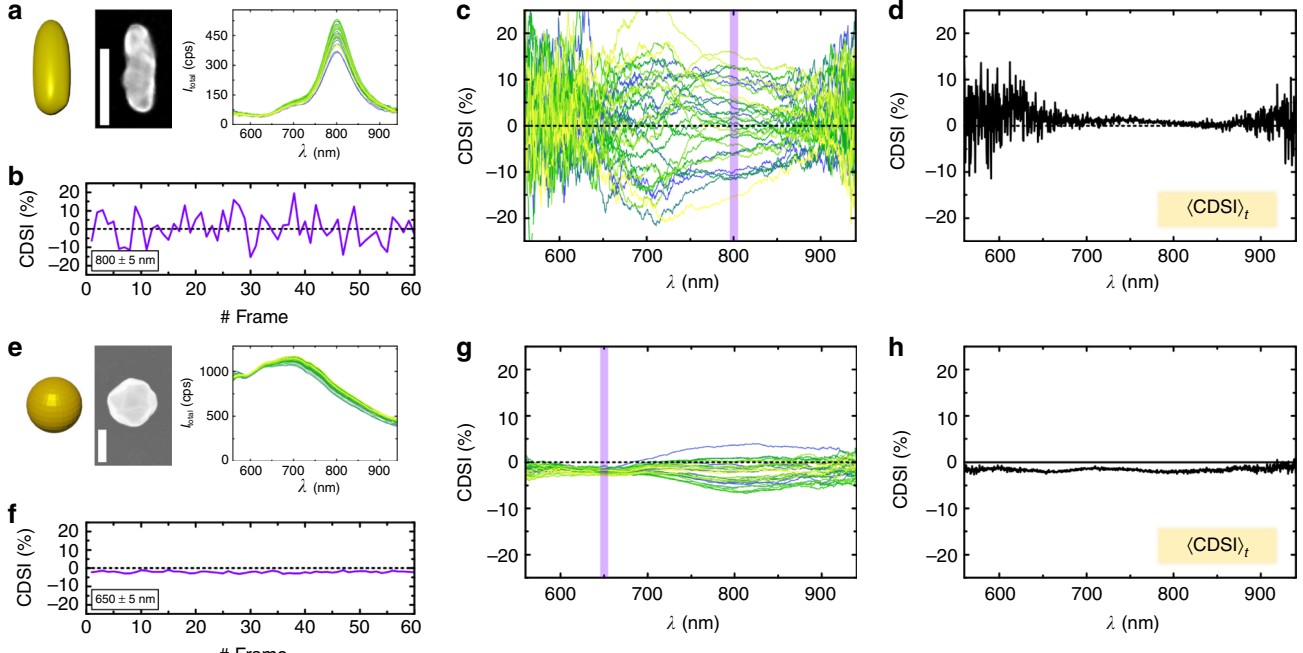

**Fig. 3 Chiroptical spectroscopy of a freely diffusing single Au nanorod and Au nanosphere. a** Schematic, SEM (scale bar 100 nm) and total scattering intensity spectra of a single Au nanorod. **b** Frame-wise CDSI signal at scattering resonance ($\lambda = 800 \pm 5$ nm) of the Au nanorod moving under random Brownian motion. **c** Full CDSI spectra of an Au nanorod, acquired subsequently and thus with different stationary orientations. The purple vertical line indicates the spectral region shown as a time-trace in (**b**). **d** $\langle CDSI \rangle_t$ measured on an Au nanorod and averaged over $t = 60$ frames from (**c**). **e** Schematic, SEM (scale bar 100 nm) and total scattering intensity spectra of a single Au nanosphere. **f** Frame-wise CDSI signal at scattering resonance ($\lambda = 650 \pm 5$ nm) of the Au nanosphere moving under random Brownian motion. **g** Full CDSI spectra of an Au nanosphere, again shown for subsequent acquisitions. Again, the purple vertical line indicates the spectral region shown as a time-trace in (**f**). **h** $\langle CDSI \rangle_t$ measured on an Au nanorod and averaged over $t = 60$ frames from (**g**).

nanorod with a peak at $\lambda = 800$ nm can be seen in Fig. 3a where the total scattering intensity $I_{total} = I_L + I_R$ is plotted. The corresponding CDSI at $\lambda = 800 \pm 5$ nm of 60 frames of a time-series taken with 1 s exposure is shown in Fig. 3b. As expected, the signal fluctuates and displays, similar to the chiral particle, nonzero CDSI spectra during the sequence. In Fig. 3c, 30 full spectra of the same measurement confirm this observation. However, the time-averaged CDSI over 60 frames (Fig. 3d) yields a zero CDSI, when a sufficiently large number of orientations have been statistically sampled. Ensemble-averaged measurements of the same sample display a zero CDSI both in our setup and in a commercial CD spectrometer (see Supplementary Fig. 3 and Note 3). Both averages are in agreement and their spectra reveal that the Au rod is an achiral nanoparticle, as expected.

The same experiment was repeated with single Au nanospheres (Fig. 3e) to demonstrate that LD and LB underlies the instantaneous CDSI signals from optically anisotropic scatterers. The observed CDSI fluctuations (Fig. 3f, g) from the 150 nm Au spheres are negligibly small, since a sphere is optically isotropic and thus a rotational motion does not change its interaction with the input light. The time-averaged CDSI of the sphere is zero, which is expected for an achiral nanoparticle (Fig. 3h).

In conclusion, chiroptical spectroscopy of complex shaped nanoparticles at fixed orientation requires knowledge of the alignment of the nanoparticle with respect to the input (and output) polarization (distribution) of the light in order to properly interpret the resulting spectra. When looking at individual spectra in time, we observe that an oriented achiral nanorod momentarily shows a nonzero CDSI, whereas the time-averaged CDSI spectrum approaches zero. Our time-averaged measurements are thus much simpler to interpret, since they are immune to imperfections in the setup and since contributions from LB and LD average to zero leaving only signals that originate from the chirality of the structure (truly chiral signals).

**Spectroscopy of a single nanohelix with external orientation control.** To validate our experimental approach and spectral analysis, we also performed experiments with a single nanostructure whose orientation could be controlled by means of a weak magnetic field. We achieved this by modifing the fabrication of the Au nanohelices to incorporate a small section of magnetic material (see "Methods"). A constant external magnetic field will thus exert a torque that will align the magnetic moment and hence the nanostructure in the direction of the external magnetic field vector (see schematic in Fig. 4a). The alignment of a single nanohelix fully concured with the corresponding spectral observation (see Supplementary Fig. 5 and Note 4).

A right-handed nanohelix was forced to follow a magnetic field that was swept along a trajectory on a unit sphere to statistically sample an isotropic distribution of orientations (see Fig. 4a). While the single nanohelix is driven through the orientation sequence we recorded its CDSI spectra. We moved the nanohelix 10 times along this path to record a total of 1000 CDSI spectra. The resultant time-averaged spectrum is shown in Fig. 4b together with 30 individual spectra randomly selected from the time-series. For comparison the time-average of the same particle undergoing free Brownian motion ($t = 400$ frames) is plotted, too. The CDSI of both time-averages $\langle CDSI \rangle_t$ agree within the accuracy of the experiment. Moreover, they agree with the previous results obtained from the freely diffusing particle experiments (Fig. 2d). It is seen that for certain orientations the CDSI signal is distorted (due to orientation and LB and LD effects). In addition, we can now also map the recorded CDSI onto the orientation of the helix's long axis. This is shown in Fig. 4c, which corresponds to the CDSI at $\lambda = 800 \pm 5$ nm of the time series in Fig. 4b. Each dot indicates a particular orientation (averaged over 10 spectra) and the color indicates the strength and polarity of the CDSI. The CDSI continuously varies along the trajectory and tends toward its maximum values at the poles, whereas the polarity changes at the equator of the sphere representing the particle's orientation. This is expected because the annular dark-field condenser illuminates the nanoparticle with a hollow light cone (Fig. 1c) and thus the helix changes its alignment with respect to the electric field vector of the incident light field when the direction of its long axis varies its $z$-coordinate (see Supplementary Fig. 5 and Note 4). The finite helix technically possesses $C_2$ symmetry, but imperfections and the magnetic handle give it $C_1$ symmetry. It is thus anticipated that the CDSI varies if the helix's long axis changes in $z$, and once more depending which end of the helix points up. On the contrary, changing the alignment in the $xy$-plane cannot be distinguished with our setup (see Supplementary Fig. 5 and Note 4). Altogether, this emphasizes again that the orientation of the nanostructure in a single-particle spectroscopy experiment plays a crucial role for the CDSI and only a statistical sample of an isotropic distribution will provide a spectrum that is a true reflection of its chirality—here the single-particle equivalent of a chiral ensemble spectrum.

## Discussion
In conclusion, we have reported the measurement of a circular differential scattering spectrum acquired from a single chiral nanoparticle that is freely suspended in a liquid. A balanced detection scheme allows us to simultaneously and continuously record the left- and right-circularly polarized scattering spectra of a single nanostructure that undergoes Brownian motion. We

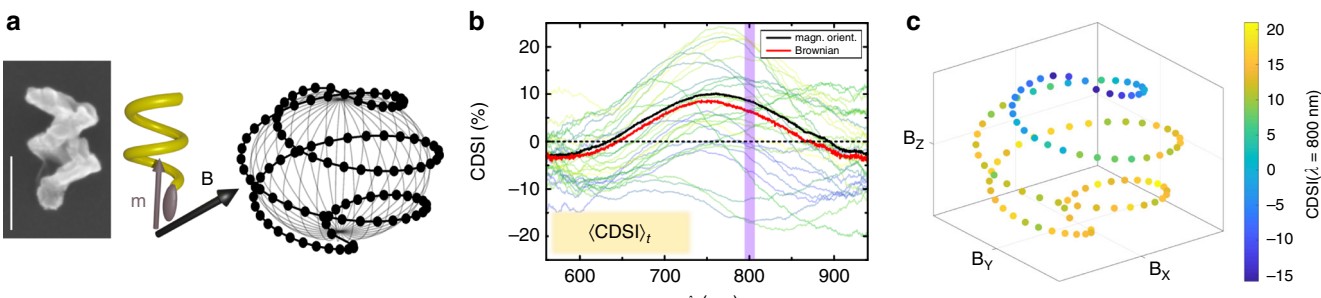

**Fig. 4 Chiroptical spectroscopy of an externally controlled single nanohelix. a** SEM (scale bar 100 nm) and schematics of an Au nanohelix with magnetic moment **m** (brown segment at end of helix) that will align with respect to an external magnetic field **B** (black). Forced sampling of isotropic orientations is achieved by evenly distributing points (black dots) on the surface of a sphere. **b** $\langle CDSI \rangle_t$ over 10 forced isotropic sampling sequences with $i = 100$ points (black) and $\langle CDSI \rangle_t$ of the same particle undergoing Brownian motion ($t = 400$ frames, red). Thirty randomly selected spectra of individual frames are depicted in the background, whereas the purple vertical line indicates the spectral region shown in (**c**). **c** CDSI at $\lambda = 800 \pm 5$ nm (10 spectra per coordinate) mapped onto the external field coordinates.

showed that the time-averaged and thereby rotationally averaged CDSI spectrum unequivocally arises from chirality intrinsic to the single particle, whereas contributions due to LB and LD cancel out. By recording the time-averaged CDSI from a single chiral nanostructure we show that it is possible to deduce the spectrum of a chiral ensemble of identical nanoparticles. This cannot be measured otherwise. We also succeeded in demonstrating the ergodic principle for chiral spectroscopy by showing the equivalence of the time-averaged chiral spectra recorded for single nanostructures with corresponding ensemble spectra. Achiral nanoparticles can momentarily also show large non-zero circular intensity difference spectra, but because the structural chirality of the achiral particle is zero, the time-averaged CDSI spectra consequently vanishes. Thus, only if the particle is chiral is the time-averaged CDSI nonzero, similar to the bulk CD spectrum recorded from a chiral ensemble. We confirm our findings and interpretation with a magneto-plasmonic nanohelix that can be oriented in solution with a magnetic field. The possibility to perform time-resolved chiroptical spectroscopy on well-controlled single chiral nanoparticles advances the field of single-particle spectroscopy and at the same time simplifies the analysis of chiral single-particle spectra. Accessing the true chiral signature of a single nanostructure in solution brings single-molecule techniques to chiroptics and may enable new forms of chiral sensing and spectroscopy.

## Methods

**Nanoparticle sample preparation.** The chiral nanohelices were grown by physical vapour deposition under an oblique angle of incidence and simultaneous rotation of the substrate around its normal[25]. Left-handed and RH two-turn helices (Au:Ti alloy, atom ratio of 15:1) were fabricated and immersed in $H_2O$ via ultrasonication of the wafer holding the nanohelices. The nanorods have been fabricated with the same technique, but with pure Au and without rotation of the substrate during deposition. The chiral nanohelices with a magnetic moment have been prepared by first growing a small Ni:Cu pillar (atom ratio 10:1) by utilizing a fast substrate rotation followed by a 2-turn Au:Ti helix as above. SEM were recorded with a Zeiss Gemini electron microscope to determine the geometric dimensions of the nanoparticles. The outer diameter of the helices is $d = 70$ nm and their height is $h = 150$ nm, while the rods have a diameter of $d = 30$ nm and a height of $h = 150$ nm. Citrate-stabilized nanospheres were purchased by Sigma Aldrich and their diameter is $d = 110$ nm. For every measurement one part of the colloidal dispersion was mixed with 20 parts glycerol before conducting the experiments in a sealed sample chamber consisting of two cover slips held by a double sided adhesive tape. The resulting viscosity of the water–glycerol mixture was $\eta \approx 370$ mPas (at 300 K). By approximating the nanohelices as a cylinder with end caps and utilizing a bead-shell model, we numerically calculated their translational and rotational diffusion constants[31]: $D_{trans} = 1.03 \times 10^{-2}$ μm$^2$ s$^{-1}$, $D_{rot}^{short} = 1.55$ rad$^2$ s$^{-1}$ and $D_{rot}^{long} = 3.64$ rad$^2$ s$^{-1}$. An exposure time of 1 s corresponds to a rotational diffusion of about $D_{rot}^{long} \approx 10$ rad$^2$ s$^{-1}$ (for 2 degrees of freedom) and thus suggests that reorientation of the nanohelices plays a role during the acquisition of one spectrum. However, the rotational diffusion is slow enough that not all orientations are sampled equally during the acquisition time. For the nanorods we calculate (with the same model): $D_{trans} = 1.86 \times 10^{-2}$ μm$^2$ s$^{-1}$, $D_{rot}^{short} = 6.94$ rad$^2$ s$^{-1}$ and $D_{rot}^{long} = 30.2$ rad$^2$ s$^{-1}$. The rotation around the rods' long axis does not alter its interaction with light, whereas a rotation around the short axis does lead to a change, but again it is not fast enough to sample a full isotropic distribution of orientations. The translational diffusion of the commercial nanospheres is $D_{trans} = 7.9 \times 0^{-3}$ μm$^2$ s$^{-1}$. Note, that a lower viscosity will lead to a much faster measurement time for a rotationally averaged chiroptical spectrum. In contrast, the translational diffusion still has to be slow enough that the particle does not leave the field of view/the focus during the course of a measurement (see Supplementary Note 1).

**Single-particle spectroscopy setup.** The spectrometer was built on a Zeiss Axio Observer inverted microscope equipped with immersion ultra condenser (NA 1.2–1.4) and a 40× (NA 0.9) EC Plan-Neofluar objective. The scattered light leaving the microscope propagates through the optical train primarily consisting of two superachromatic QWPs, a 1° WP and apertures to avoid cross-talk between the LCP and RCP light. The first QWP converts the scattered circularly polarized light into two orthogonal linear polarizations, which are separated in the WP. The spatially separated beams are then sent through a second QWP in order to convert the linearly polarized light back to circularly polarized light. This is to avoid differences in intensity throughput of linearly polarized components by the mirrors and the diffraction grating of the Czerny–Turner-spectrograph (Andor Shamrock

193i, with a 300 lines/mm silver-coated grating). The spatially separate beams are projected onto an open-electrode CCD detector (Andor DU920P-OE), which was chosen to avoid fringing in the near-infrared. The wavelength resolution of the detector is 0.415 nm/px. The setup was calibrated by utilizing 1 μm $TiO_2$ spheres (in 1:20 water:glycerol) as a spectrally flat reference scatterer allowing us to deconvolve the lamp spectrum and setup efficiency from our single particle spectra. To show several spectra from individual frames in one plot we had to smooth the raw spectra over intervals of adjacent wavelengths (9.13 or 22.8 nm) as otherwise the data would be obscured by noise. More details about the acquisition and evaluation workflow for single-particle spectra is given in Supplementary Note 1.

For controlling the orientation of the magnetic nanostructures in 3D, a custom setup with three magnetic Helmholtz coil pairs was mounted on the microscope stage with the sample in its centre. The field strength was 3.8 mT. A custom written Python code controlled the field vector via an analog output card and synchronised the magnetic field motion pattern with the shutter and exposure timing of the spectrometer.

**Ensemble spectroscopy measurements.** To retrieve the isotropic ensemble averaged CDSI, a suspension of the particles was measured with all apertures on our setup opened as far as possible to allow scattered light from a large area within the sample to reach the detector. By taking the mean of 60 detector rows, the field of view we are averaging over corresponds to ~40 μm × 20 μm in the sample plane. Thus tens of particles contributed to each frame, which was recorded with an exposure time of 3 s. A sequence of 400 frames was acquired, which corresponds to a total measurement duration of 20 min, thus ensuring ensemble averaging. The number density of the nanoparticle suspension was on the order of $N/V \approx 10^9$ mL$^{-1}$ after they have been immersed in $H_2O$ via ultrasonication. Measurements with commercial citrate stabilized Au nanoparticles in $H_2O$ were also performed with number densities of $N/V \approx 10^9$ mL$^{-1}$. The classical ensemble CD measurements took place in $H_2O$ in a four-side polished cuvette using a commercial CD spectrometer (JASCO J-810) that had its detector mounted in a 90° rotated alignment with respect to the incident light direction (see Supplementary Note 2).

## Data availability

All the findings of this study are available in the main text or the Supplementary Information. The raw data are available from the corresponding author upon reasonable request.

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

## Acknowledgements
We thank H.-H. Jeong, G. Markovich, and U. Hananel for helpful comments and suggestions, C. Miksch for help with the sample preparation, and J. Spatz for scanning electron microscope access. This work was in part supported by the Max Planck-EPFL Center for Molecular Nanoscience and Technology.

## Author contributions
J.-P.G. and A.M. planned the optical layout. J.-P.G. constructed the spectrometer and J.S. the magnetic coil setup. J.-P.G. and J.S. designed the experimental workflow in equal parts. J.S. prepared the samples, wrote the control and evaluation routine (MATLAB) and conducted all the single particle measurements. P.F. supervised and directed the research project. J.S., J.-P.G., and P.F. wrote the paper.

## Funding

## Competing interests
The authors declare no competing interests.
