## [Peer Review File · Nature Communications]

REVIEWER COMMENTS

Reviewer #1 (Remarks to the Author):

This manuscript reports circular differential scattering intensity (CDSI) for “single” nano particles, using a laudable idea: namely, time averaged CDSI spectrum of a freely diffusing nano particle is equivalent to the ensemble average spectrum of nano particles. This idea avoids the complications from orientational effects (linear dichroism, quadrupole contribution etc) of a oriented particle. The authors are able to not only measure the time averaged CDSI spectra of a single nano helix, but also show that the time averaged CDSI spectra of a single achiral sphere becomes zero. The authors are to be commended for having taken strenuous efforts to demonstrate that the observed effects are real.

Clarifications for the following few questions will be useful.

(1). The authors mentioned that the solution used is made of one part colloidal dispersion and 20 parts glycerol. How would one make sure that this dilution is sufficient for only one particle diffuses into the field of view of microscope objective ?

(2). Regarding the statement in the Supporting Information,

“The column dimension of the detector is used for spectral separation. Since we used a 40x objective, each row on the CCD corresponds to a rectangular area with a width of 650 nm in the sample plane. The size of the nanostructures being measured is on the order of $d \approx 150$ nm. It is thus reasonable to assume that only a few adjacent rows on the detector are capturing the spectrum of the particle IP article(λ). The other detector rows will collect light that is scattered by particles as well as impurities in the vicinity and hence they are used for the background correction.”

(a). Is it not possible to observe multiple scattering peaks from different particles across the 650 nm width of CCD ?

(b). It is not clear how the scattering from other “particles as well as impurities” serves as background ? Would that not amount to another peak from another particle?

Reviewer #2 (Remarks to the Author):

This manuscript represents a significant milestone in our emerging ability to probe the fundamental properties of isolated supramolecular structures by lifting the veil of statistical (ensemble) averaging

that has long obscured all manifestations of heterogeneous phenomena. More specifically, the authors have employed an ingenious time-resolved method to perform linear chiroptical spectroscopy on individual plasmonic nanoparticles of prescribed chiral or achiral morphologies as they freely circulate (via random Brownian motion) through a viscous solution-phase environment. By simultaneously discriminating the left-circular and right-circular components of scattered light following unpolarized excitation in a dark-field microscope, temporal “snapshots” of wavelength-resolved circular differential response for single particles could be acquired continuously. These data encode effects of both intrinsic chirality (circular dichroism and birefringence) and extrinsic orientation (linear dichroism and birefringence) for the diffusing target, with the cumulative spectrum obtained by averaging a series of such “snapshots” serving to cancel orientational “artifacts” and reveal (for what may be the very first time) the true chiroptical signatures of an isolated plasmonic entity. To further validate their approach, the authors repeated measurements with magnetic nanostructures that enable alignment and circulation to be controlled by an external magnetic field, thereby highlighting the pronounced dependence of optical activity on the spatial orientation of a particle. The results obtained during this tour de force study are intriguing and informative, promising to elicit a wide range of subsequent research endeavors designed to elaborate the nature of plasmonic chiroptical processes as well as their potential applications in various forms of chiral sensing/measurement.

The contents of this manuscript are eminently suitable for publication as a report in Nature Communications, with key results being presented in a logical and concise manner that should be accessible to the general scientific community. Prior to immediate publication, the authors should consider clarification of the following minor points:

1. At the bottom of page 4, the authors state that “spectra of a single particle were acquired at one specific instant in time during which it is nearly stationary and thus has a defined orientation in 3D space.” Might it be possible to quantify the phrases “nearly stationary” and “defined orientation” given that the time needed to record a complete chiroptical spectrum appears to be on the order of 1 second? How much translation and rotation (on average) will the targeted nanoparticles undergo during this acquisition time? A brief discussion of these issues does appear in the “Methods” section and in the Supplementary Information, so perhaps it would be sufficient to point the reader there for further details. It also would be useful to mention how long a single nanoparticle can be tracked using the current implementation of the apparatus, as this ultimately will place limits on spectral-averaging procedures.

2. It would be most informative if the circular-differential spectra for individual nanoparticles (such as those in Fig. 2) could be placed on an absolute intensity scale akin to the circular-differential absorption (ellipticity) or anisotropy ratio metrics routinely employed for ensemble-averaged studies. This may not be possible given the nature of the measured signals (per text on page 7), but it would highlight the exceptionally large magnitude of chiroptical effects supported by plasmonic nanoparticles. Perhaps the definition of CDSI given in equation (1) already affords a reasonable

counterpart for the conventional anisotropy ratio, although the instructive discussion of chiral scattering processes given in the Supplementary Information (Section 2) would tend to make this assertion a bit too simplistic.

3. In the sentence spanning pages 7 and 8 of the manuscript the authors refer to “achiral electric-quadrupolar terms” responsible for circular differential intensities that “do not reflect the chirality intrinsic to the molecule.” This statement needs to be clarified, as it may be a bit misleading in present form. Presumably these achiral effects are distinct from the chiral electric-quadrupole (E2) and magnetic-dipole (M1) interactions that separately interfere with accompanying electric-dipole (E1) interactions to yield conventional linear chiroptical signatures? In fact, the resulting E1-E2 and E1-M1 processes are believed to contribute equally to the overall chiroptical response of an oriented molecule, yet the former rotationally average to zero and thus vanish in an isotropic ensemble average.

4. Were any experimental attempts made to quantify effects of the environment (for example, the nature of the solvent, as well as its temperature and/or viscosity) on measurements of the response evoked from entrained nanoparticles? Presumably, changes in any solvent parameter could (at least) affect the orientational dynamics of a nanoparticle, thus placing bounds on the extent of temporal-snapshot averaging needed to yield true chiroptical signatures.

5. Conventional circular-differential spectra (as measured for an isotropic ensemble of chiral targets) stem from the trace of a second-rank chiroptical tensor, the individual elements of which contain invaluable information regarding subtle matter-field interactions. In particular, the relatively small value of the trace often reflects the partial cancellation of sizable positive and negative contributions that relate the spatial properties of intrinsic transition moments to those of applied electromagnetic fields. Provided that the orientation of a targeted chiral particle was known (perhaps by exploiting the novel external-field control paradigms described by the authors), it might be possible to extract the entire tensor by analyzing each of the temporal snapshots individually. Although this could prove to be very difficult to accomplish in practice (given potential artifacts arising from linear dichroism/birefringence and annular dark-field illumination), it would provide access to unique information that typically is available only for chiral species constrained under strongly perturbative conditions (for example, immobilized on a surface or imbedded in a crystal). Any comments along these lines would make a tantalizing addition to the manuscript, especially since the results highlighted in Fig. 4c might already be partially on the way to realizing this possibility!

6. The authors are encouraged to perform a critical proof reading of their manuscript in order to correct grammatical errors and improve the overall quality of presentation. For example, the final sentence in the topmost paragraph on page 3 appears to be a fragment (perhaps due to a missing comma?). More importantly, the various colored curves appearing in Fig. 2c should be identified fully in the caption (the legend really is not clear here; there seem to be two nearly identical curves

for each particle). The caption also indicates that the solid and dashed curves in Fig. 2e correspond to bulk CD spectra measured at different scattering angles, but no further details are given. Finally, the first line of the caption for Fig. 3 appears to have inverted the set of panels corresponding to a gold nanosphere (should be panels (a-d)) and a gold nanorod (should be panels (e-h)).

We thank both reviewers for their comments. We are delighted to receive positive feedback on our work. In what follows we reprint the initial comments in black, answer in blue and indicate the new text and changes in the paper in red.

Reviewer #1 (Remarks to the Author):

This manuscript reports circular differential scattering intensity (CDSI) for "single" nano particles, using a laudable idea: namely, time averaged CDSI spectrum of a freely diffusing nano particle is equivalent to the ensemble average spectrum of nano particles. This idea avoids the complications from orientational effects (linear dichroism, quadrupole contribution etc) of a oriented particle. The authors are able to not only measure the time averaged CDSI spectra of a single nano helix, but also show that the time averaged CDSI spectra of a single achiral sphere becomes zero. The authors are to be commended for having taken strenuous efforts to demonstrate that the observed effects are real.

We thank the referee for the positive feedback.

Clarifications for the following few questions will be useful.

(1). The authors mentioned that the solution used is made of one part colloidal dispersion and 20 parts glycerol. How would one make sure that this dilution is sufficient for only one particle diffuses into the field of view of microscope objective ?

The aqueous colloidal solution was sufficiently diluted so that the particles were usually well separated. If particles were too close to each other or too bright (outliers, aggregates) they were disregarded. Before acquisition of a time-series one promising particle was located and the stage moved to center it in the field of view. Single particles can be distinguished from aggregates due to the scattering intensity. To address the comment, we revised the first paragraph of section 1 in the SI and added an additional figure for clarification:

"The column dimension of the detector is used for spectral separation. Since we used a 40x objective, each row on the CCD corresponds to a rectangular area with a width of 650 nm in the sample plane. The size of the nanostructures is on the order of $d \approx 150$ nm. It is thus reasonable to assume that only a few adjacent rows on the detector capture the spectrum of a single particle. The other detector rows will collect light stemming from other scatterers in the field of view as well as impurities in the vicinity. Such effects contribute to a pale intensity distribution, which is used for a background correction (see SI-Fig XXX). Prior to a spectral acquisition, an individual particle was selected and centered by moving the microscope stage. Single particles could be

distinguished from agglomerates due to the vastly differing scattering intensities. Multiple particles in the field of view were readily identified by the corresponding peaks registered at different positions on the CCD.”

SI-Fig XXX Snapshot of the CCD showing the raw signal. Particles can be distinguished from the low intensity background due to stronger localized intensities. Spatially separated particles appear on different rows of the CCD. This permits the identification of a scatterer of interest. The particle concentration is kept dilute enough to ensure that the chance of several particles entering the field of view is very low.

(2). Regarding the statement in the Supporting Information,

“The column dimension of the detector is used for spectral separation. Since we used a 40x objective, each row on the CCD corresponds to a rectangular area with a width of 650 nm in the sample plane. The size of the nanostructures being measured is on the order of $d \approx 150$ nm. It is thus reasonable to assume that only a few adjacent rows on the detector are capturing the spectrum of the particle IP article(λ). The other detector rows will collect light that is scattered by particles as well as impurities in the vicinity and hence they are used for the background correction.”

(a). Is it not possible to observe multiple scattering peaks from different particles across the 650 nm width of CCD ?

In general this is possible, but additional particles would show up as intense signals on different rows on the CCD. If the particles come too close to each other, they cannot be

distinguished as individual particles any more. However, the particles show translational diffusion over the course of the measurement and even if the particles would be stacked above each other at the start of the experiment, they appear on different rows later (see also the answer to question 1/the new figure SI-Fig XXX).

(b). It is not clear how the scattering from other "particles as well as impurities" serves as background ? Would that not amount to another peak from another particle?

Yes, if a single strong scatterer in focus enters the spectrum then it appears as a peak. However, the many defocused particles and other scattering impurities lead to a low intensity constant background (see also the answer to question 1/the new figure SI-Fig XXX).

Reviewer #2 (Remarks to the Author):

This manuscript represents a significant milestone in our emerging ability to probe the fundamental properties of isolated supramolecular structures by lifting the veil of statistical (ensemble) averaging that has long obscured all manifestations of heterogeneous phenomena. More specifically, the authors have employed an ingenious time-resolved method to perform linear chiroptical spectroscopy on individual plasmonic nanoparticles of prescribed chiral or achiral morphologies as they freely circulate (via random Brownian motion) through a viscous solution-phase environment. By simultaneously discriminating the left-circular and right-circular components of scattered light following unpolarized excitation in a dark-field microscope, temporal "snapshots" of wavelength-resolved circular differential response for single particles could be acquired continuously. These data encode effects of both intrinsic chirality (circular dichroism and birefringence) and extrinsic orientation

(linear dichroism and birefringence) for the diffusing target, with the cumulative spectrum obtained by averaging a series of such "snapshots" serving to cancel orientational "artifacts" and reveal (for what may be the very first time) the true chiroptical signatures of an isolated plasmonic entity. To further validate their approach, the authors repeated measurements with magnetic nanostructures that enable alignment and circulation to be controlled by an external magnetic field, thereby highlighting the pronounced dependence of optical activity on the spatial orientation of a particle. The results obtained during this tour de force study are intriguing and informative, promising to elicit a wide range of subsequent research endeavors designed to elaborate the nature of plasmonic chiroptical processes as well as their potential applications in various forms of chiral sensing/measurement.

The contents of this manuscript are eminently suitable for publication as a report in Nature Communications, with key results being presented in a logical and concise manner that should be accessible to the general scientific community. Prior to immediate publication, the authors should consider clarification of the following minor points:

We thank the referee for these very positive comments and for recommending publication of our article in Nature Communications.

1. At the bottom of page 4, the authors state that "spectra of a single particle were acquired at one specific instant in time during which it is nearly stationary and thus has a defined orientation in 3D space." Might it be possible to quantify the phrases "nearly stationary" and "defined orientation" given that the time needed to record a complete chiroptical spectrum appears to be on the order of 1 second? How much translation and rotation (on average) will the targeted nanoparticles undergo during this acquisition time? A brief discussion of these issues does appear in the "Methods" section and in the Supplementary Information, so perhaps it would be sufficient to point the reader there for further details. It also would be useful to mention how long a single nanoparticle can be tracked using the current implementation of the apparatus, as this ultimately will place limits on spectral-averaging procedures.

The last paragraph on the bottom of page 4 should give the reader a general idea of the respective times. However, we extended the paragraph and included the following:

"In our setup, a spectrum of a single particle was acquired at one specific instant in time, i.e. with an exposure time short enough to assume a quasi-stationary orientation in 3D space during the acquisition of one spectrum (see Methods and SI). If the particle moves and if the observation is repeated ..."

In addition, we added a sentence in the Methods section (at the end of the "Nanoparticle Sample Preparation" paragraph) to point out the limits of our setup:

"Note, that a lower viscosity will lead to a much faster measurement time for a rotationally averaged chiroptical spectrum. In contrast, the translational diffusion still has to be slow enough that the particle does not leave the field of view/the focus during the course of a measurement (see SI)."

In the SI (first section, beginning of second paragraph) we added the following:

"Because the nanoparticles can freely diffuse, the maximum measurement time t_{\max} for different particles and viscosities can be estimated with the Brownian mean square displacement. The dimensions of the aperture set a spatial constraint $\Delta x = 5\mu\text{m}$ in the object plane: $\langle(\Delta x)^2\rangle = 2 D t_{\max}$. Hence, a 150nm diameter nanosphere can be observed for approximately 25 minutes (1:20 glycerol), whereas in water ($\nu \approx 1\text{mPas}$) it will leave the observation volume after approximately 4 seconds. In general, the nanoparticles can change their position (row on the CCD) over time between subsequent acquisitions."

2. It would be most informative if the circular-differential spectra for individual nanoparticles (such as those in Fig. 2)

could be placed on an absolute intensity scale akin to the circular-differential absorption (ellipticity) or anisotropy ratio metrics routinely employed for ensemble-averaged studies. This may not be possible given the nature of the measured signals (per text on page 7), but it would highlight the exceptionally large magnitude of chiroptical effects supported by plasmonic nanoparticles. Perhaps the definition of CDSI given in equation (1) already affords a reasonable counterpart for the conventional anisotropy ratio, although the instructive discussion of chiral scattering processes given in the Supplementary Information (Section 2) would tend to make this assertion a bit too simplistic.

Typically, single-particle spectra are normalized to the total scattering intensity in order to permit comparison with other measurements. In our opinion, it is problematic to quote an ellipticity (θ) or an anisotropy (g -factor) for a single particle. Ellipticities typically consider the sample concentration, which is challenging to be specified for a single particle measurement, since the observation volume is hard to determine in practice. The g -factor is based on a difference in extinction (which is also depended on concentration), whereas our setup only probes the scattering and not the absorption of a particle. We thus think it is better to avoid any potential confusion and we would not want to introduce a new quantity in a field that already has several established measures of optical activity.

3. In the sentence spanning pages 7 and 8 of the manuscript the authors refer to "achiral electric-quadrupolar terms" responsible for circular differential intensities that "do not reflect the chirality intrinsic to the molecule." This statement needs to be clarified, as it may be a bit misleading in present form. Presumably these achiral effects are distinct from the chiral electric-quadrupole (E2) and magnetic-dipole (M1) interactions that separately interfere with accompanying electric-dipole (E1) interactions to yield conventional linear chiroptical signatures? In fact, the resulting E1-E2 and E1-M1 processes are believed to contribute equally to the overall chiroptical response of an oriented molecule, yet the former rotationally average to zero and thus vanish in an isotropic ensemble average.

We completely agree with this comment and what we meant is exactly what the referee describes. We rephrased the following in order to avoid any confusion:

"For molecules it is well known from the theory of optical activity [17] and the measurement of circular dichroism [20] that an oriented sample, including molecules immobilized at an interface, displays spectra that significantly differ from rotationally-

averaged (ensemble) CD spectra. In the latter case, contributions from electric-quadrupole transitions average to zero, whereas those transitions have to be considered for a stationary aligned structure. Similarly, contributions due to any linear birefringence (LB) or linear dichroism (LD) in an (oriented) sample can alter the polarization state and contribute to the circular intensity difference signal [18], which does not reflect the chirality intrinsic to the molecule.”

4. Were any experimental attempts made to quantify effects of the environment (for example, the nature of the solvent, as well as its temperature and/or viscosity) on measurements of the response evoked from entrained nanoparticles? Presumably, changes in any solvent parameter could (at least) affect the orientational dynamics of a nanoparticle, thus placing bounds on the extent of temporal-snapshot averaging needed to yield true chiroptical signatures.

We considered such influences, in particular whether heating could locally affect the viscosity, but these effects were deemed to be small. The environmental parameters (temperature, viscosity) are relatively constant over the course of the measurement as can be seen by looking at the time-traces of the CDSI signals (e.g. Fig. 2b or Fig. 3a). The rate of the fluctuations and their extent does not change. One could also expect that heating due to the light would scale with the input light intensity, but we did not observe this. Changes of different solvent viscosities are also discussed in the response to question 1.

5. Conventional circular-differential spectra (as measured for an isotropic ensemble of chiral targets) stem from the trace of a second-rank chiroptical tensor, the individual elements of which contain invaluable information regarding subtle matter-field interactions. In particular, the relatively small value of the trace often reflects the partial cancellation of sizable positive and negative contributions that relate the spatial properties of intrinsic transition moments to those of applied electromagnetic fields. Provided that the orientation of a targeted chiral particle was known (perhaps by exploiting the novel external-field control paradigms described by the authors), it might be possible to extract the entire tensor by analyzing each of the temporal snapshots individually. Although this could prove to be very difficult to accomplish in practice (given potential artifacts arising from linear dichroism/birefringence and annular dark-field illumination), it would provide access to unique information that typically is available only for chiral species constrained under strongly

perturbative conditions (for example, immobilized on a surface or imbedded in a crystal). Any comments along these lines would make a tantalizing addition to the manuscript, especially since the results highlighted in Fig. 4c might already be partially on the way to realizing this possibility!

The reviewer highlights a fascinating possibility. There are several effects that need to be dis-entangled in recording (and interpreting) orientation-resolved spectra and they do not seem trivial. We have considered this, but feel that this needs more careful analysis before we can attempt such a measurement. For instance, the helix can still rotate about its own body axis (about the magnetic moment) and so this will cause some orientational freedom, which would need to be considered in a separate average as the coordinate frame of the particle rotates with respect to that of the incident light. Connecting these individual measurements back to a theoretical model are also very challenging as the scattering occurs from a range of angles into a separate set of angles (segment of the scattering cone). So, while fascinating we do not (yet) feel confident to attempt this.

6. The authors are encouraged to perform a critical proof reading of their manuscript in order to correct grammatical errors and improve the overall quality of presentation. For example, the final sentence in the topmost paragraph on page 3 appears to be a fragment (perhaps due to a missing comma?). More importantly, the various colored curves appearing in Fig. 2c should be identified fully in the caption (the legend really is not clear here; there seem to be two nearly identical curves for each particle). The caption also indicates that the solid and dashed curves in Fig. 2e correspond to bulk CD spectra measured at different scattering angles, but no further details are given. Finally, the first line of the caption for Fig. 3 appears to have inverted the set of panels corresponding to a gold nanosphere (should be panels (a-d)) and a gold nanorod (should be panels (e-h)).

We thank the referee for the suggestions. We have carefully re-read the manuscript and fixed smaller grammatical issues/wordings, and in addition we changed the following clarifications:

Caption Fig.2:

“Time-resolved chiroptical spectroscopy of freely diffusing single nanohelices. a) SEM and schematics of Au-nanohelices (scale bar 100 nm). b) Frame-wise CDSI at $\lambda = 750 \pm 5$ nm for a single left-handed nanohelix that is undergoing Brownian motion. c) Full CDSI spectra for momentarily aligned orientations in 3D space (first 30 frames shown). d) CDSI ($t=400$ frames) for freely diffusing single Au-nanohelices and –spheres. **For each shape two measurements from different particles are shown in different shades of**

grey, red and blue. e) CDSI of the same Au-helices and -spheres presented in (d) measured in the dark-field setup according to Fig.1b (solid lines) and by utilizing a classical chiroptical measurement with a cuvette (dashed lines, data scaled). The latter detects under a different scattering angle (see SI).”

Caption Fig. 3:

“Chiroptical spectroscopy of a freely diffusing single Au nanorod (a-d) and Au nanosphere (e-f). a) Schematic, SEM (scale bar 100 nm) and total scattering intensity spectra of a single Au nanorod. b) CDSI signal at scattering resonance ($\lambda = 800 \pm 5$ nm) of the Au nanorod moving under random Brownian motion. c) CDSI spectra corresponding to b). d) CDSI averaged over 60 frames from c). e) Schematic, SEM (scale bar 100 nm) and total scattering intensity spectra of a single Au nanosphere. f) CDSI signal at scattering resonance ($\lambda = 650 \pm 5$ nm) of the Au nanosphere moving under random Brownian motion. g) CDSI spectra corresponding to f). h) CDSI averaged over 60 frames from g).”

REVIEWERS' COMMENTS:

Reviewer #1 (Remarks to the Author):

The changes made by the authors appear to adequately address the reviewer comments

Reviewer #2 (Remarks to the Author):

The authors have done a comprehensive and quite commendable job of addressing all of the substantive issues raised by this referee during the initial review of their manuscript. As such, this impressive work is now deemed suitable for immediate publication as a report in Nature Communications.

REVIEWERS' COMMENTS:

Reviewer #1 (Remarks to the Author):

The changes made by the authors appear to adequately address the reviewer comments

Reviewer #2 (Remarks to the Author):

The authors have done a comprehensive and quite commendable job of addressing all of the substantive issues raised by this referee during the initial review of their manuscript. As such, this impressive work is now deemed suitable for immediate publication as a report in Nature Communications.

*** We want to thank again both reviewers and appreciate their work, which certainly helped to improve the manuscript.